# Ratings of Hand Activity and Force Levels among Women and Men Who Perform Identical Hand-Intensive Work Tasks

**DOI:** 10.3390/ijerph192416706

**Published:** 2022-12-13

**Authors:** Gunilla Dahlgren, Per Liv, Fredrik Öhberg, Lisbeth Slunga Järvholm, Mikael Forsman, Börje Rehn

**Affiliations:** 1Section of Sustainable Health, Department of Public Health and Clinical Medicine, Umeå University, S-901 87 Umeå, Sweden; 2Radiation Physics, Department of Radiation Sciences, Umeå University, S-901 87 Umeå, Sweden; 3IMM Institute of Environmental Medicine, Karolinska Institutet, S-171 77 Stockholm, Sweden; 4Division of Ergonomics, School of Engineering Sciences in Chemistry, Biotechnology and Health, KTH Royal Institute of Technology, Hälsovägen 11C, S-141 57 Huddinge, Sweden; 5Department of Community Medicine and Rehabilitation, Physiotherapy, Umeå University, S-901 87 Umeå, Sweden

**Keywords:** gender differences, equity, repetitive strain injury, cumulative trauma disorders, upper extremity, ergonomic assessment, workload, exposure assessment, observation, psychophysics

## Abstract

We compared hand activity and force ratings in women and men doing identical hand-intensive work tasks. Musculoskeletal disorders are more common in women and hand-intensive work leads to an increased risk of these disorders. Knowledge of the gender influence in the rating of work exposure is lacking. The aim of this study was to investigate whether women and men performing identical hand-intensive work tasks were equally rated using hand activity and normalized peak force levels with the Hand Activity Threshold Limit Value^®^. Fifty-six workers participated, comprising 28 women–men pairs. Four observers—two woman–man pairs—were also involved. Self-ratings and observers’ ratings of hand activity and force level were collected. The results of these ratings showed no significant gender differences in self-rated hand activity and force, as well as observer-rated hand activity. However, there was a significant gender difference in the observer-rated force, where the women were rated higher (mean (SD): women 3.9 (2.7), men 3.1 (1.8) (*p* = 0.01)). This difference remained significant in the adjusted model (*p* = 0.04) with grip strength and forearm–finger anthropometrics. The results provide new insights that observers’ estimates of force can be higher in women compared with men in the same work tasks. Force should be further investigated and preferably compared to objective measurements.

## 1. Introduction

Risk assessment in hand-intensive work is of great value for detecting the risk of musculoskeletal disorders (MSDs) in the neck and upper extremities. A commonly used method for the assessment of hand activity and force is the Hand Activity Threshold Limit Value^®^ (HA TLV^®^). It is aimed at evaluating the exposure of the wrist and forearm to risk [1,2] and showing the risk levels for carpal tunnel syndrome [3]. It is not yet known whether women and men are rated equally when using this method. We need to know whether they are assessed equally to best protect both women and men workers from work-related ill-health, such as MSDs. As the first step in the occupational health and safety risk management process, knowledge of adequate and correct ratings for both sexes is essential. 

MSDs due to work entail large costs for individuals and society. Industries with a higher prevalence of MSDs are affected in terms of lost productivity due to employees’ days away from work [4]. Roughly three out of every five workers have reported MSD complaints during the previous 12 months [4]. There are differences between men and women regarding MSDs. The prevalence of MSDs is higher in women compared with men [4,5,6]. Of all female workers with a work-related health problem, 60% identified MSDs as their most serious issue [5]. Female workers show a higher seven-day prevalence of MSDs in the elbow/hand (27%) compared with men (19%), and a doubled risk for the a diagnosis of carpal tunnel syndrome (CTS) in the wrist [7]. 

Sectors that reported a high incidence of MSDs in the upper extremities, and specifically CTS, are the meat- and fish-processing industries, industrial production, dentistry, hairdressing, cleaning [6] and assembly [6,8]. Exposure factors at work associated with CTS that were described in systematic reviews (Habib et al., van Rijn et al., Palmer et al.) [8,9,10] and a meta-analysis (You et al.) [11] are the use of vibrating tools and prolonged work with repetitive flexion and extension of the wrist, especially when combined with forceful grip or pressure [8,9,10,11]. However, these studies lack stratification by sex. The prevalence of CTS and other MSDs in the arm and hand, as well as the relation to physical work exposure and sex, is described somewhat differently across the literature. A similar risk for MSDs and CTS in women and men with similar work exposure was described by Nordander et al. [6]. Nordander et al.’s cross-sectional study showed that the risk of CTS and other MSDs in the hand–arm was elevated in repetitive/constrained work compared with varied/mobile work. Further, the risk was similarly elevated for females and males [6]. A systematic review by Bellini et al. showed that women in surgery experienced more musculoskeletal pain than men [12]. For interventionists, Barnard et al. reported that women physicians experienced more work-related MSDs in the upper extremities compared with men [13]. In a cross-sectional study of automobile assembly-line workers by Fransson et al., the risk for musculoskeletal symptoms in the forearm–hand was increased in women, and their exposure to repetitive movements, precision movements and manual handling (<15 kg) was higher when compared with men [14]. The measures of exposure duration in these studies ranged from one workday to several years. 

Strength and anthropometric factors differ between the genders. Men are generally stronger than women and hand grip strength is no exception. When grip strength peaks between the ages of 25–39, women have, on average, 65% of men’s grip strength [15], and this difference remains throughout life [15,16]. Furthermore, the anthropometric body size is, on average, larger for men compared with women [17,18,19]. Studies also showed that there is a correlation between grip strength and the circumference of the hand [20], and between grip strength and forearm circumference and length, hand size and body height [21]. In real life, the external workload in an identical repetitive work task situation is in absolute terms the same for women and men. It can, however, be assumed that women work with higher relative intensity in relation to their lower maximum strength and size. This was confirmed in a study testing muscular activity in two industries with female and male workers and identical repetitive tasks, where females had substantially higher relative muscular activity and increased MSD prevalence in the upper extremity compared with men [22]. Due to the average differences between women and men in grip strength and body size, it might be reasonable to assume that the perceived force level would be higher for women and lower for men in identical hand-intensive work tasks.

The HA TLV^®^ is used among practitioners in occupational health, such as ergonomists and occupational hygienists, and in companies [1,2]. It is based on ratings using two psychometric scales: hand activity level regarding hand exertions/repetitions and estimation of normalized peak force for the hand. This method is considered time- and cost-effective. It is not yet known whether women and men are rated equally in self- versus observer-assessed hand activity and hand force level when performing an identical hand-intensive work task. Moreover, the influence of grip strength and hand and arm anthropometric size has not been established. This knowledge would be important in the MSD risk assessment of women and men using observational methods. Measurements of women and men workers performing single identical hand-intensive work tasks would enable us to focus exclusively on physical exposure. Measurements over a longer period at work might also reflect gender differences in terms of other factors, such as work organization and type of work tasks. 

Thus, the aim of this study was two-fold: first, to investigate whether women and men performing an identical hand-intensive work task are equally assessed in self and observer ratings of hand activity and force demand levels; second, to investigate whether any gender differences in the ratings of hand activity and force are related to grip strength, anthropometrics of the forearm and finger abduction.

## 2. Materials and Methods

### 2.1. Study Sites and Participants

Fourteen companies in Sweden that involved manual hand-intensive work tasks in various sectors were contacted by phone (by G.D.). The aim was to have a diversity of companies and work tasks with different intensities regarding hand repetition and force to best capture a wide range of physical exposure levels. First, the employers were informed about the study and asked for permission to assess risk exposure in employees in pairs of a woman and a man performing an identical hand-intensive work task. Eight companies agreed to participate in the study and signed a letter of intent. The companies who declined to participate gave the following reasons: major ongoing organizational changes (n = 3), only women workers (n = 1), not hand-intensive work (n = 1) and the COVID-19 pandemic (n = 1). The recruited companies were in the following sectors: warehouse work, pharmaceutical production, industrial assembly work, postal service delivery, postal sorting terminal work and postal sorting of direct mail, manual packaging of portion-packed food, and laboratory analysis and pipetting. Inclusion criteria for the worker participants were performing hand-intensive tasks at work and being able to work without difficulty with their arms rated <1 on the following scale: 0 = no difficulty, 1 = some difficulty, 2 = a lot of difficulty and 3 = unable to work. This scale was inspired and adapted using the Work Activity Limitation Scale [23]. Sixty-seven participants fulfilled these criteria and volunteered to participate. After the exclusion of 11 worker participants (not hand-intensive work n = 2, simulated work tasks n = 2, lack of a pair of a woman and man n = 3, illness n = 4), 56 participants, i.e., 28 pairs of one woman and one man, remained in the final study population.

### 2.2. Work Tasks

At the initial information meetings, participants and employers were asked to identify common hand-intensive tasks performed equally by a woman and a man for >4 h per day. They were encouraged to apply the selection criteria collaboratively. The selected work tasks were then discussed with the researcher. In total, 28 unique pairs of a woman and a man in 18 different (unique) work tasks were identified (Table 1). 

### 2.3. Procedure

At the information meetings, workers who agreed to participate in the study received a composite questionnaire. Approximately one week later, the participant met individually with the researcher (G.D.) in a quiet room at the workplace and was investigated according to the assessor’s protocol (grip strength, anthropometrics and a physical examination). On a later occasion, in most cases within 0–7 days, the participant was observed and video recorded while performing the chosen hand-intensive work task. Finally, directly after the execution of the work task, the participant self-rated the hand activity and force. All assessments were administered by G.D., who is a registered physiotherapist and specialist in ergonomics and orthopedics and experienced in the clinical assessment of patients and occupational risk assessment of workplace exposures.

### 2.4. Composite Questionnaire

The participating workers answered a questionnaire about work experience (work hours/day), illness, sick leave (Work Ability Index) [24], stress (time pressure and general stress according to Quick Exposure Check) [25] and physical activity (IPAQ-SF) [26,27].

### 2.5. Musculoskeletal Complaints and Clinical Examination

The Nordic Questionnaire was used for the assessment of complaints [28], and a standardized systematic clinical evaluation, namely, the Health Evaluation in Adverse Conditions (HECO), was used to assess pain prevalence in the neck, shoulder, arm and hand, as well as criteria for diagnoses from these areas [29,30].

### 2.6. Strength and Anthropometrics

Some anthropometric measures that should differ between men and women and are suitable for field on-site studies at various companies were chosen. Grip strength was tested with a JAMAR Plus+ Digital Dynamometer (Patterson Medical, China, Medema) for both the right and left hand in a standardized position sitting comfortably on a stool without an armrest while holding the tested arm at 90 degrees flexion with the elbow comfortably close to the body, the forearm straight forward, neutral wrist and semi-pronated hand with the thumb facing upward and gripping the handle with all fingers at the second handle position [31]. The participant was instructed to do one comfortable test grip to ensure familiarization and a correct position, followed by one maximal grip; standardized verbal encouragement was given during the trial to motivate the participant to exert his/her maximal grip strength. The JAMAR was calibrated before the recording period.

Body height was measured using a folding rule taped to the wall (cm). Body weight was measured in kilograms with a body scale (Coline^®^ 34-5062, serial no 100749399). Forearm length was measured while sitting with the elbow flexed at 90 degrees, semi-pronated forearm resting on the table, and hand and fingers aligned straight forward. The distance from the back of the olecranon to the tip of digit three was measured in centimeters. Finger abduction, which is a dynamic measure, with the fingers and thumb stretched as widely apart as the person found comfortable, was measured from the outer border of the tip of digit five to the outer border tip of digit one in centimeters [19]. 

### 2.7. Self-Ratings of Hand Activity and Force

The psychometric scales of hand activity and normalized peak force (force) were self-rated by the worker in a quiet room to exclude the potential influence of other workers. Hand activity was rated with the hand activity level scale (0–10) to assess hand exertions, rest pauses and speed of motion (0 = hands idle most of the time with no regular exertions; 10 = rapid, steady motion/difficulty keeping up or continuous exertion) [1]. The normalized peak force was assessed using a Borg CR-10 [32]. The force rating was the perception of load of relative level of effort on a scale from 0 to 10 that a person of average strength would exert in the same posture required by the task. This scale consisted of the following values and anchors: 0 (nothing at all); 0.5 (very, very easy); 1 (very easy); 2 (easy); 3 (moderately hard); 4 (somewhat hard); 5 (hard); 6 and 7 (very hard); and 8, 9 and 10 (very, very hard). The researcher (G.D.) explained both scales orally and in writing, with the scales given on a laminated sheet for each participant. The participants rated the hand activity and normalized peak force on the most active hand, normally the dominant hand (n = 55), in their regular work for familiarization before the work task, and then directly after execution of the targeted work task. 

### 2.8. Video Recordings

All participants were video-recorded (by B.R.) performing the hand-intensive work task. They were instructed to work for 15 min, which is a common length of time used for observing individual hand-intensive work tasks. The participant was filmed so that the face, upper body, hands and arms, and whole body were captured dynamically during the task.

### 2.9. Observers’ Ratings of Hand Activity and Force

The same ratings of hand activity and force were also made by experienced observers. For this, four ergonomists who were also registered physiotherapists (women (n = 2), men (n = 2)) were contacted by phone, informed about the study and invited to participate as experienced observers (henceforth called “observers”). Their work experience ranged from 15 to 35 years. Two observer teams were formed comprising one woman and one man to reduce gender effects in the observer assessment. In a one-hour session, they were informed and trained about rating the hand activity level and force. In a quiet room, in two sessions of four hours, each team made a collective assessment of the hand activity and force level scales of each worker from the video of the face, upper body, hands, arms and whole body. To avoid the teams influencing each other, they rated on separate occasions. Each team assessed half (n = 28, 14 pairs of participants) of the total population (n = 56). They assessed video recordings (15 min) of each participant performing the work task. They were given a sheet with the hand activity level scale and Borg CR-10 scale for force and were instructed to assess these two parameters jointly. The order of the videos was mixed, and they were free to choose the order themselves within the set time. The observers were not informed about each worker’s biological gender, nor the pairing of women and men in the study’s analysis of workers executing identical hand-intensive work tasks; they were asked only to focus on the rating of each participant. However, from the videos, it was apparent whether it was a woman or a man who performed the task.

### 2.10. Analyses

A power analysis showed that to detect a difference in the HA TLVs^®^ between women and men, assuming an effect size (Cohen’s d) of 0.8 and a significance level of 5%, group sizes of 27 women and 27 men were required. That is, a total of 54 people was needed to achieve 80% statistical power. The hand activity level and the Borg scales were treated as continuous variables [32]. Normal distribution for the variables was investigated using visual assessments from histograms and Q–Q plots. For comparing mean exposure levels between women and men, a paired samples t-test was used. This was done for both the self and observer ratings. A linear mixed model was used to compare the exposure levels between women and men, adjusted for right-hand grip strength, forearm length and finger abduction size. The model was as follows:$$y_{ij}=\beta_{0}+\beta_{1}x_{1ij}+\beta_{2}x_{2ij}+\beta_{3}x_{3ij}+\beta_{4}x_{4ij}+\alpha_{i}$$
*i* = 1, 2, …, 28, *j* = 1, 2,
where *y_ij_* is the exposure assessment of the jth worker in the ith pair. *β*_0_, *β*_1_, *β*_2_, *β*_3_ and *β*_4_ are fixed effects of the intercept, gender, hand grip strength, forearm length and finger abduction size, respectively. Further, *x*_1*ij*_, *x*_2*ij*_, *x*_3*ij*_ and *x*_4*ij*_ represent the gender (binary variable), hand grip strength, forearm length and finger abduction width, respectively, of the jth worker in the ith pair. Finally, *α_i_* is a random effect due to differences in exposure between the pairs on the ith pair under an assumption of a variance component covariance structure. The parameter of interest was *β*_1_, which represented the difference in assessments between men and women. The significance level was set at 0.05 in all analyses. 

## 3. Results

### 3.1. Description of the Workers

Workers’ self-reported data and measures regarding demography, complaints, diagnoses, sick leave, physical activity, work exposure and work stress are presented in Table 2.

### 3.2. Measures of Grip Strength, Forearm Length and Finger Abduction

For hand grip strength and anthropometric measures of the right forearm and finger abduction, there were statistically significantly smaller values for women compared with men (Table 3). 

### 3.3. Distribution of Self- and Observer-Rated Hand Activity and Force Levels in the Work Tasks

The distribution of self- and observer-ratings on a 0–10 scale for women and men workers is presented in Table 4. 

### 3.4. Unadjusted Comparison of Hand Activity and Force

In the unadjusted analysis, the paired t-tests did not show statistically significant differences between women and men in the mean self-rated hand activity (Table 5). 

### 3.5. Linear Mixed Model Analysis

There was a significant difference with regard to the force ratings made by the observers (*p* = 0.01, Table 5). This difference remained significant (*p* = 0.04) when adjusting for right-hand grip strength, forearm length and finger abduction. 

## 4. Discussion

The main results showed no significant differences between women and men in the self-rated hand activity and force, nor the observer-rated hand activity level. However, the observers rated force significantly higher in women compared with men who performed identical hand-intensive work tasks. This difference in observer-rated force also remained statistically significant after adjusting for grip strength, forearm length and finger abduction.

Comparable studies investigated hand activity and force ratings in identical real work tasks, and data was presented on hand activity and force as measured by the Hand Activity Threshold Limit Value^®^ in terms of the reliability (test–retest, inter-rated reliability) [1,33], validity (compared with a rating with the Strain Index) [33], methods of execution (on-site versus off-site) [34] and in populations in relation to CTS [3,33,35,36]. However, these studies lacked a comparison between women and men.

In our study, the women were, on average, younger, had shorter work experience, lower seven-day pain prevalence for the neck–shoulder and hand–arm, and fewer diagnoses of the elbow–hand. However, these women had a higher number of diagnoses related to the neck and shoulder compared with the men, which agreed with the reported statistics [4]. Some of the gender differences in complaints and diagnoses might have been due to workstations being adapted for a higher body height, thereby increasing adverse neck and shoulder postures in women. As expected, the women in our study had lower grip strength and anthropometric measures (forearm, finger abduction) in comparison to the men. Women’s grip strength was, on average, 60.1% of the men’s, and the women’s forearm length and finger abduction were 90.5% and 89.6% of the men’s, respectively. This was in accordance with reference data on grip strength [15] and anthropometrics [17,18,19]. 

The men’s self-rated hand activity and force were somewhat higher compared with the women’s, but there were no statistically significant differences. This was interesting since we expected that due to men being stronger and bigger, they would have a lower perceived exertion of force. However, our results were in accordance with findings by Slopeci [37] and Srinivasan [38], who studied time to fatigue (rated as Borg ≥ 8) in a repetitive pointing task at shoulder height. Neither study found any significant difference in the time to fatigue between women and men. 

While observers did not rate the hand activity of women and men significantly differently in our study, they did rate the hand force of women higher and men were rated lower when performing identical hand-intensive work tasks for both the unadjusted and adjusted results for the impact of grip strength and forearm–hand size. This means that even though we considered a comparison between women and men with similar anthropometric features, the difference remained. This difference may have been caused by gender bias due to the men being taller and having bigger upper limbs compared with the women, which may be unconsciously perceived as a need for lower force for men compared with women in the identical work task. There may also be unconscious expectations of increased MSDs vulnerability in women. Wurzelbacher [34] reported similar findings for ratings of hand activity and force in different sectors (hospital/outpatient laboratory, bus manufacturing plant, engine assembly plant). For hand activity in their study, there was a substantial agreement between the two compared methods, i.e., observer rating on site (direct) versus off site (video-based). For the hand force observed using an on-site rating compared with worker self-rating, they reported a moderate agreement. Furthermore, Lowe and Krieg [39] compared observer ratings by 29 ergonomists of five workers and six manual work tasks (internal validity), which showed that ergonomists were able to estimate temporal aspects, such as the duration and frequency of forceful exertions (hand activity) more precisely than the magnitude of the exertions (force). However, neither of these two studies was stratified by the gender of the worker nor the observer. Both studies indicated that hand activity is somewhat more accurate compared with force, similar to our results. The results of the current study raised the question of biological differences regarding which force level is the “true” force value in women and men. Therefore, future research should focus on validity in estimating the force level in women and men during hand-intensive activities at work.

### 4.1. Methodological Considerations

A strength of this study was the novel research design that compared the perception of physical exposure levels in hand activity and force, both self- and observer-rated, in women and men performing an identical hand-intensive work task. Knowledge about possible gender differences is valuable for research and clinical practice when assessing hand activity and force in women and men. Further, the use of a single hand-intensive work task executed on site during a real and identical hand-intensive work situation by both a woman and a man enabled focusing on the actual perceived physical exposure. We wanted to eliminate non-physical factors that may influence the risk of MSDs in women compared with men, such as employment conditions, how work is organized, stress due to psychosocial demands, discrimination at work and segregation into jobs with higher MSD risks [4]. Therefore, unlike other studies [6,14,40] that measured over one working day or more, this study did not involve the risk of being influenced by, e.g., how work is organized for women and men. Although the number of included tasks was limited (n = 18), the assessed levels varied from low to high values for hand activity and from low to medium values for power in both women and men, reflecting a similar variation in load assumed in this study of hand-intensive work.

The HA TLV^®^ method is based on subjective assessments. Observational methods generally have low reliability in comparison with objective technical measurements [41,42]. Furthermore, observations of hands, especially rapid hand movements, have lower reliability than observations of larger body regions. With today’s developments, there are easy-to-use technical methods for research and field studies. In the future, the use of such methods [43], together with scientifically based action levels [44], should be considered. The current finding of worker gender-related differences in estimates is another reason to consider the use of objective methods. 

The study was designed to detect an effect size corresponding to Cohen’s d of 0.8 with an 80% target power. Any differences smaller than this had a marked risk of going undetected due to type 1 error. Ergonomists usually observe and video-record the performance of work tasks in the actual workplace. They often have additional information about the manual force requirements of the work (such as weight of objects and push/pull forces), and sometimes also perform the work task themselves, which adds a personal experience of the force magnitude that may increase the accuracy of the force rating. In this study, the observers did not have this additional on-site information. To compensate for this, they were free to watch the videos in the order of their choosing and repeatedly during the data collection meetings. 

### 4.2. Practical Applications

These results regarding women’s and men’s perception of exertion in terms of rated activity and force in hand-intensive work are important for researchers and practitioners that assess the risk of MSDs. The fact that hand force assessed by observers was differently rated in women and men performing the same task is important to note. This shows a need to improve the precision of observers’ ratings of women and men. Since this study was a comparison of the estimation of women’s and men’s efforts, these results tell us nothing about what the true exposure was and who was closest to the true exposure value. This indicated the need for validation of these estimates while taking gender into account. A reliable estimate of the exposure at work will improve the understanding of differences in MSDs in women and men. For occupational health professionals that assess the risk of MSDs, it is important to be aware of the risk of observers overestimating the risk for women compared with men, which may reflect a gender bias in observers. Awareness of gender differences in the assessment of hand activity and force is an important key issue in the occupational health and safety risk management process toward sustainability, good health, high productivity, and longer working life for both women and men.

## 5. Conclusions

This study showed no significant differences between women and men in self-rated hand activity and force, nor in observer-rated hand activity level. However, the observers rated the hand force significantly higher in women compared with men who performed identical hand-intensive work tasks. This difference remained after adjusting for grip strength, forearm length and finger abduction. The results indicated a systematic gender difference in observers’ force ratings and the addition of more objective methods may therefore be required. 

## Figures and Tables

**Table 1 ijerph-19-16706-t001:** The identified work tasks with various hand-intensive exposures. Each pair consisted of a woman and a man who performed the identical hand-intensive work task. There were 28 pairs in total.

Work Task	Work Task Description	Pairs, n
1	Ranking of goods	2
2	Picking, base products, heavier load	2
3	Picking, fruit, vegetables, lighter load	1
4	Cassette filling	2
5	Manual decontamination of bags	2
6	Inspection, labeling, packaging of ampoules	1
7	Fluid inspections of bottles	1
8	Hose winding	1
9	Hose coupling	1
10	Small parts picking, scanning	1
11	Wheeling	1
12	Paternoster picking	1
13	Manual sorting of mail	4
14	Manual sorting of cataloges	2
15	Steamplicity, manual packaging of food portions	2
16	Manual sorting of direct mail	2
17	Manual pipetting	1
18	Water filtration	1

**Table 2 ijerph-19-16706-t002:** Description of the workers.

	Women	Men
Demography, anthropometrics and lifestyle	n = 28	n = 28
Age, years	33.2 (12.1) ^1^	36.8 (12.3) ^1^
Dominant hand	Right n = 23	Right n = 26
	Left n = 4	Left n = 2
	Bilateral n = 1	Bilateral n = 0
Smoking	No n = 26	No n = 28
	Yes n = 2	Yes n = 0
Body weight, kg	72.1 (15.2) ^1^	93.7 (16.1) ^1^
Body height, cm	169.2 (8.0) ^1^	182.7 (8.5) ^1^
BMI	24.1 [21.6, 27.6] ^2^	27.3 [25.5, 30.9] ^2^
Complaints [28]		
Pain in the neck or shoulders the last 7 days	n = 13, 46.4%	n = 16, 57.1%
Pain in the elbow or hands the last 7 days	n = 9, 32.1%	n = 11, 39.3%
Diagnoses from the neck and shoulder [29,30]	n = 11	n = 2
For women/men: tension neck syndrome n = 1/n = 1, cervicalgia n = 1/0, thoracic outlet syndrome n = 1/0, acromioclavicular syndrome n = 3/1, biceps tendinitis n = 4/0 and supraspinatus tendinitis n = 1/0		
Diagnoses from the hand and arm [29,30]	n = 4	n = 7
For women/men; De Quervain n = 2/0, overused hand syndrome n = 1/0, pronator teres syndrome n = 0/1, carpal tunnel syndrome n = 0/2 and ulnar nerve entrapment elbow n = 0/1		
Sick leave [24]		
Number of days the last year	1.9 (1.1) ^1^	2.1 (1.1) ^1^
0 days	13	9
1–7 days	7	11
8–24 days	6	5
25–99 days	1	2
100–365 days	1	1
Physical activity last 7 days; all activities including work, transport, housework, gardening, leisure activities and planned exercise [26,27]		
Number of days with vigorous physical activity >10 min	3.0 (2.0) ^1^	3.0 (2.5) ^1^
Average time (hours) per day with vigorous physical activity	1.0 [0.6, 1.1] ^2^	1.0 [0.7, 1.6] ^2^
Number of days with moderate physical activity >10 min	3.4 (2.1) ^1^	3.5 (2.0) ^1^
Average time (hours) per day with moderate physical activity	0.7 [0.5, 3.0] ^2^	1.5 [1.0, 3.0] ^2^
Number of days walking >10 min	7.0 [6.0, 7.0] ^2^	7.0 [4.0, 7.0] ^2^
Average time (hours) per day walking	1.0 [0.5, 2.0] ^2^	1.0 [0.5, 1.0] ^2^
Time per day sitting (hours)	4.1 [3.5, 7.0] ^2^	4.0 [3.0, 6.0] ^2^
Work exposure		
How many years of working experience do you have with hand-intensive tasks?	5.7 [1.5, 13.5] ^2^	11.0 [2.4, 22.7] ^2^
How many hours per day do you work during a normal day with hand-intensive tasks, repeated movements and exertions?	5.4 (2.0) ^1^	5.3 (1.8) ^1^
How many hours per day do you work during an intensive day with hand-intensive tasks, repeated movements and exertions?	6.0 [5.5, 8.0] ^2^	7.0 [6.0, 8.0] ^2^
Stress [25]		
Do you have difficulty keeping up with this work?		
Never	n = 5 (17.9%)	n = 5 (17.9%)
Sometimes	n = 20 (71.4%)	n = 22 (78.6%)
Often	n = 3 (10.7%)	n = 1 (3.6%)
In general, how do you find this job?		
Not at all stressful	n = 3 (10.7%)	n= (14.3%)
Mildly stressful	n = 18 (64.3%)	n = 16 (57.1%)
Moderately stressful	n = 5 (17.9%)	n = 8 (28.6%)
Very stressful	n = 2 (7.1%)	n = 0 (0%)

^1^ Mean (SD), ^2^ median, interquartile range.

**Table 3 ijerph-19-16706-t003:** Right-hand grip strength, forearm length and finger abduction comparison between women and men.

Variables	Women *	Men *	*p*-Value **
Right grip, JAMAR, kg	35.5 (6.8)	58.7 (10.0)	<0.001
Right forearm length, cm	43.9 (2.1)	48.5 (2.2)	<0.001
Right finger abduction, cm	19.8 (1.3)	22.1 (1.6)	<0.001

* Mean (SD), ** paired samples test.

**Table 4 ijerph-19-16706-t004:** Self- and observer-rated hand activity and force in women and men in the 18 work tasks with numbers (n) of women and men represented in three levels (0–3, 4–6 and 7–10) of hand activity and force.

	Women and Men		Ratings 0–3		Ratings 4–6		Ratings 7–10	
Variables	Min	Max	Women, n	Men, n	Women, n	Men, n	Women, n	Men, n
Hand activity self-rated	1	8	3	1	18	17	7	10
Force self-rated	0.5	9	21	16	7	12	0	0
Hand activity observer	1	8	7	8	12	13	9	7
Force observer	0.5	9	13	16	10	12	5	0

**Table 5 ijerph-19-16706-t005:** Ratings as the means and the unadjusted (paired samples *t*-test) and adjusted (linear mixed model) differences for the covariates grip strength, forearm length and finger abduction.

	Women ^1^	Men ^1^	Unadjusted		Adjusted	
			Diff [95% CI]	*p*-Value ^2^	Diff [95% CI]	*p*-Value ^3^
Hand activity self-rating	5.6 (1.6)	6.2 (1.4)	−0.6 [−1.22, 0.04]	0.07	0.4 [−0.98, 1.77]	0.57
Force self-rating	3.1 (1.4)	3.3 (1.4)	−0.2 [−0.89, 0,50]	0.57	0.2 [−1.23, 1.54]	0.82
Hand activity observer	5.0 (1.9)	4.9 (2.0)	0.1 [0.57, 0.79]	0.75	−0.1 [−1.84, 1.61]	0.90
Force observer	3.9 (2.7)	3.1 (1.8)	0.8 [0.26, 1,42]	0.01	1.7 [0.05, 3.29]	0.04

^1^ Mean (SD), ^2^ paired samples *t*-test, ^3^ linear mixed model.

## Data Availability

Not applicable.

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
