# Peer review of "Ratings of Hand Activity and Force Levels among Women and Men Who Perform Identical Hand-Intensive Work Tasks"

_ijerph, 2022, doi:10.3390/ijerph192416706_

Round 1
Reviewer 1 Report
This is a very skillfully designed study, with methods described in great detail enabling repetition by other researchers if needed. Matching the pairs is almost complete - a very rare case among respective studies.
All the aspects are well-covered in the discussion; the only issue, which may be further discussed is the question of the extent to which the hand grip force measurement reflects the forces needed in the tasks in concern in the real life (particularly in view of more common neck-shoulder diagnoses of the female workers that may affect the use of force in the forearm but not hand grip?).
Do you have any recommendations for occupational health practitioners?
Author Response
Please see the attachement.
Kind regards,
Mrs Gunilla Dahlgren

Reviewer 2 Report
See attached file

Author Response
Please see the attachement.
Kind regards
Mrs Gunilla Dahlgren

Round 2
Reviewer 2 Report
Authors have answered my concerns and nuanced their findings.